

# Market-approved convolutional neural network tasked with classifying skin lesions under suspicion of melanoma: performance across primary care clinics within Australia

Ian J. Miller[1], Michael Stapelberg[1,2], Jeremy Hudson[1,3], Paul Coxon[1,3], Nathaniel Milani[3], Nedeljka Rosic[1,4], James Furness[5], Joe Walsh[6,7] and Mike Climstein[8]

[1] Aquatic Based Research, Faculty of Health, Southern Cross University, Bilinga, Qld, Australia
[2] Skin Clinic Robina, Robina, Qld, Australia
[3] North Queensland Skin Centre, Unaffiliated, Townsville, Qld, Australia
[4] Department of Biomedical Sciences, Faculty of Health, Southern Cross University, Bilinga, Qld, Australia
[5] Water Based Research Unit, Bond University, Robina, Qld, Australia
[6] Sport Science Institute, Sydney, NSW, Australia
[7] AI Consulting Group, Sydney, NSW, Australia
[8] Physical Activity, Lifestyle, Ageing and Wellbeing Faculty Research Group, University of Sydney, Sydney, NSW, Australia

Corresponding author
Ian J. Miller,
i.miller.11@student.scu.edu.au

## ABSTRACT

**Background.** Artificial intelligence (AI) is poised to revolutionise how melanoma is detected in clinical practice, yet few studies have been published with patient data at the forefront.

**Objective.** The primary aim of this study was to investigate the clinical performance of a market-approved convolutional neural network (CNN) to better differentiate skin lesions suspicious of being malignant melanoma (MM). A secondary aim of this study was to compare the diagnostic performance of the CNN across two separate general practices, that are skin cancer focused clinics.

**Methods.** Multicentre, cross-sectional study using a commercially available CNN on 373 melanocytic lesions (114 melanoma, 259 non-melanoma) from participants attending a skin examination within two Australian specialised, general practice clinics. Performance metrics included sensitivity, specificity, predictive values, diagnostic odds ratios, accuracy and area under the curve (AUC) of receiver operating characteristics (ROC) used for classification of images.

**Results.** The CNN average sensitivity [Gold Coast *vs* Townsville] was calculated as 63.2% [61.5% *vs* 68.6%], specificity as 53.9% [52.5% *vs* 55.1%], positive predictive value as 37.8% [28.9% *vs* 44.0%] and negative predictive value as 76.8% [71.4% *vs* 84.2%]. Likelihood ratios were 1.4 for positive likelihood ratio, 0.7 for negative likelihood ratio and a diagnostic odds ratio of 2.0 across both clinics. Accuracy was calculated as 56.6% [56.1% *vs* 57.5%] and the AUC of ROC for both clinics was 0.602 and 0.615 for Townsville and Gold Coast, respectively.

**Conclusions.** Improvement of the performance of this CNN for the classification of images, particularly when suspecting MM is necessary before it may be used in a clinical

setting in Australia. Other validated AI systems used internationally may also require review for use in an Australian setting.

## INTRODUCTION

Worldwide, malignant melanoma (MM) accounts for nearly one in five cancers of the skin and incidence is anticipated to increase by 50% by the year 2040 (*Arnold et al., 2022*). The highest global incidence of MM remains in Australia, with the standardised rate of 57 cases identified in every 100,000 people in the general population (*Australian Institute of Health and Welfare, 2023*). However, the standardised rate in Australian surfers has been reported to be 119.8 times higher (*Miller et al., 2023*). Detecting MM early is associated with lower morbidity and mortality and ultimately better patient outcomes (*Melarkode et al., 2023*). Conversely, detecting MM at an advanced stage, is often associated with a poorer patient prognosis and increased risk of metastatic disease (*Sandru et al., 2014*). In clinical practice, to biopsy a skin lesion suspicious for MM, the recommendation by the Royal Australian College of General Practitioners (RACGP) is to perform an excisional biopsy sampling the entire lesion and some adjacent normal appearing skin, which is then sent for histopathology assessment and reporting (*Thompson, Scolyer & Kefford, 2012*). An argument against MM screening programs is that they may result in high biopsy rates or overdiagnosis by the removal of early MM which may not progress to an invasive or life-threatening stage (*Whiteman et al., 2022*). It is, however, not currently possible to differentiate between MM that may progress and those that will remain indolent.

Artificial intelligence (AI), in particular, convolutional neural networks (CNN), has been proposed to assist clinicians with the early identification of MM, with the aim of reducing associated morbidity and mortality (*Melarkode et al., 2023*). Promising findings have been reported with 96.7% of clinically relevant melanocytic lesions being detected successfully by AI total body mapping systems in clinical settings (*Winkler et al., 2024*). Many studies investigating AI and skin cancer detection remain in the realm of computer science. These studies often test AI performance on skin challenge datasets such as the International Skin Imaging Collaboration (ISIC) (*Codella et al., 2018*) and the Human Against Machine (HAM10000) dataset (*Tschandl, Rosendahl & Kittler, 2018*). Of the studies investigating melanoma classification on market-approved CNN from patient data, the results show a great deal of heterogeneity, with accuracy reported between 44.0% and 92.0% on MM (*Miller et al., 2024*). Improved performance has been reported in European studies (*Thomas et al., 2023*; *Winkler et al., 2023*), while two studies from Australia have reported less favourable AI tool accuracy (*Miller et al., 2023*; *Menzies et al., 2023*). The Australian-based studies reported a sensitivity between 16.4–53.3%, a specificity of 54.4–98.3% and an accuracy of 54.2–80.2% for the detection of MM using AI technology (*Miller et al., 2023*; *Menzies et al., 2023*). These values show a great deal of heterogeneity, which does not
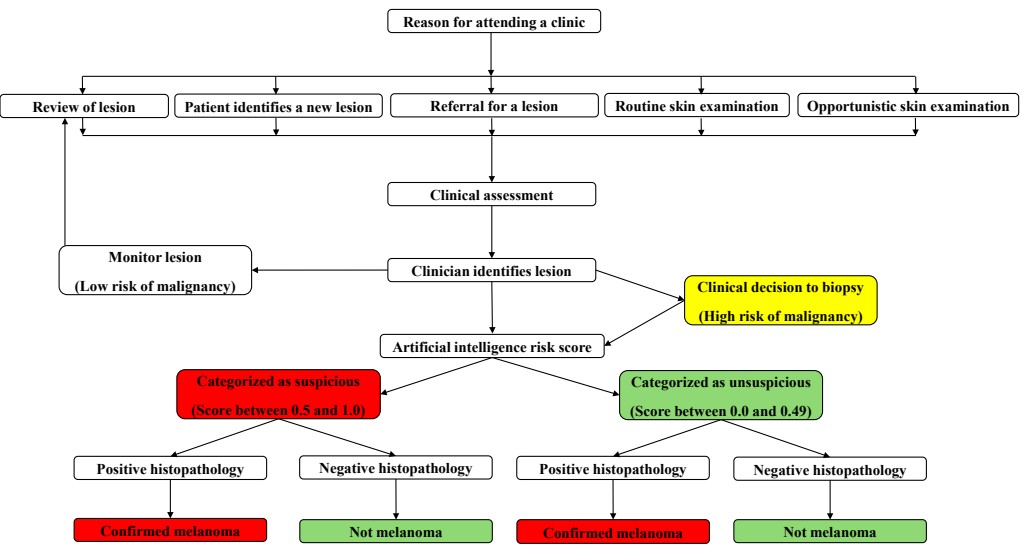

**Figure 1** Concept map of clinician workflow when examining the skin for detection of suspicious lesions.

provide confidence in relying only upon AI as a clinical or training tool. With the highest incidence of MM globally, Australia should be viewed as the benchmark for assessing AI performance in lesions suspected of MM.

The primary aim of this study was to investigate and report on the clinical performance of a market-approved CNN used at the bedside, to better differentiate lesions suspicious of MM from benign lesions. Furthermore, this study also set out to compare the diagnostic performance of the same market-approved bedside CNN used in two separate general practice, skin cancer focused clinics across Queensland (Australia).

# MATERIALS AND METHODS

## Clinical procedure and data collection

The medical practitioners who participated in this study were specialist general practitioners (GPs) with a focus on skin cancer and melanoma diagnosis and treatment. All GPs are fellows of the Royal Australian College of General Practitioners (FRACGP) and are accredited by the Skin Cancer College of Australasia (SCCA). They are expert dermoscopists with over 10 years' experience.

Prior to data collection, all GPs across both locations used a standardised protocol to assess skin lesions suspicious for MM (Fig. 1). Initially, a detailed participant history was obtained, including participant age, gender, skin type, sunburn history, naevi count, past medical history, current medications and a personal and family history of skin cancer. Then, clinicians performed a whole-body skin examination complete with dermatoscopic assessment of skin lesions.

When a skin lesion suspicious for MM was detected using dermatoscopy, a high-definition image was captured of the lesion using a bedside digital dermatoscope with
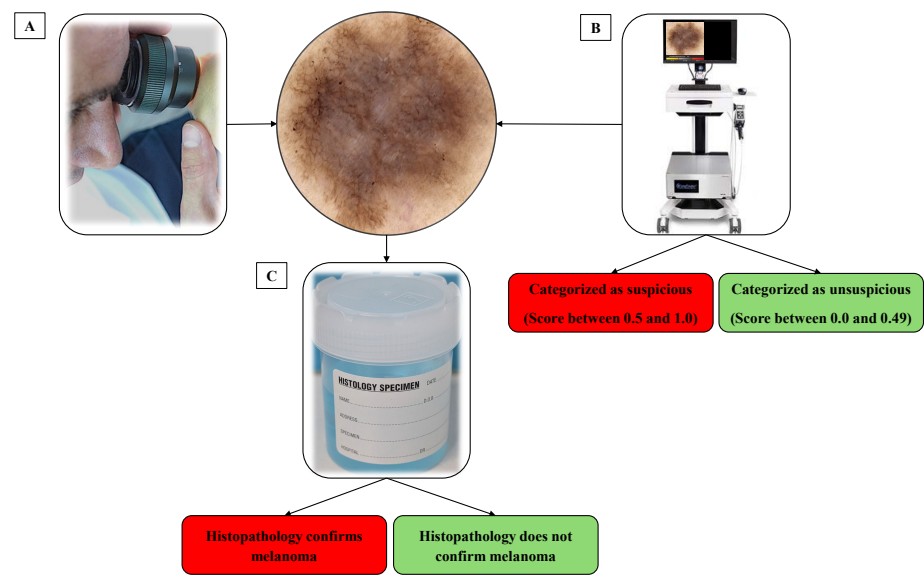

**Figure 2** **Process of identifying and classifying lesions suspect of malignancy.** (A) The clinician identifies a lesion that they suspect may be a melanoma. (B) Before completing a biopsy on the lesion, an image was captured on a market-approved dermatoscopic unit. A digital score assisted by CNN categorizes the lesion as suspicious or unsuspicious. (C) A biopsy of the lesion was taken and sent for histological analysis to determine malignancy.

a built-in market-approved CNN (Moleanalyzer-Pro; FotoFinder Systems GmbH, Bad Birnbach, Germany). The software version at the time of analysis was 3.4.1.0-(x64). A biopsy of the suspicious skin lesion was then performed. This was done using excisional saucerisation of the entire lesion with additional normal appearing surrounding skin or excisional biopsy with a two mm margin. The biopsy sample was then sent for histopathological examination by a pathologist and assessment with immunohistochemistry at a commercial laboratory to determine if the lesion was a MM, as illustrated in Fig. 2. Histopathological examination was evaluated by multiple commercial laboratories and the providers of each laboratory were accredited dermatopathologists.

Upon request, the CNN revealed a score ranging from 0 to 1.0, which was recorded and attached to the high-definition image of the skin lesion. In this study, the AI score was requested and recorded post-consultation to ensure the CNN had no influence on the clinician's workflow.

## Study design and participating population

This study was approved by the Southern Cross University's Human Research Ethics Committee (11th May 2020/47). Written informed consent was obtained from each participant after one of the researchers informed them of the details of the study and that any participation would be voluntary. This study was a multi-centre, cross-sectional diagnostic accuracy study within Queensland, Australia. This study follows the STROBE statement for cross-sectional studies. Participants were recruited from two clinics; one located in the Gold Coast region and the other in Townsville. The inclusion criteria

**Table 1 Outcome performance metrics.**

| Outcome measure | Formula |
|---|---|
| Sensitivity/True Positive Rate (TPR) | TP/(TP+FN) |
| Specificity/True Negative Rate (TNR) | TN/(FP+TN) |
| Positive Predictive Value (PPV) | TP/(TP+FP) |
| Negative Predictive Value (NPV) | TN/(FN+TN) |
| Accuracy | (TP+TN)/(TP+FP+FN+TN) |
| Prevalence | (TP+FN)/(TP+FP+FN+TN) |
| False Positive Rate (FPR) | FP/(FP+TN) |
| False Negative Rate (FNR) | FN/(FN+TN) |
| Positive Likelihood ratio (LR+) | TPR/FPR |
| Negative Likelihood ratio (LR-) | FNR/TNR |
| Diagnostic Odds ratio (DOR) | LR+/LR- |

**Notes.**
Where:
TP, True positive; TN, True negative; FP, False positivie; FN, False negative.

were participants aged 18 years or older presenting to a clinic for a skin examination. Participants were recruited between September 2021 and August 2024. Both Gold Coast and Townsville participants were individuals attending a clinic for skin cancer screening. However, it should be noted that a media campaign based in the Gold Coast area promoted the study with a focus on recruiting individuals with a history of sporting or recreational activity outdoors. Many of those who attended the Gold Coast clinic came from surfing backgrounds, which has previously been reported to have up to 120 times the melanoma prevalence (*Miller et al., 2023*) compared to the reported Australian general population. With regard to participants from Townsville there was no media campaign promoting the study. However, many of the participants provided feedback to the clinicians during the skin screening consult of a history of outdoor recreation, mainly being walking, gardening or running.

## Statistical analysis

Histopathology is the gold standard methodology of confirming malignancy in tissue samples, and in our study, the dermatopathologist's report either confirmed the diagnosis of MM or not. Confirmation of MM or not for each lesion was then associated with the image score obtained from the bedside market-approved CNN, where an AI score $\geq 0.50$ was categorised by the manufacturer as suspicious of malignancy, and an AI score $\leq 0.49$ was classed as benign. An AI value $\geq 0.50$ and a positive histopathology result represented a true positive (TP). A value $\geq 0.50$ and a negative histopathology report represented a false positive (FP). A value of negative histopathology and a value $\leq 0.49$ was a true negative (TN). A value $\leq 0.49$ and a positive histopathology report was a false negative (FN). The performance metrics (TP, FP, FN, TN) as represented in Fig. 3, were then used to calculate the equations in Table 1.

All analyses were performed using Excel (Microsoft Office 365; Microsoft Corporation, Redmond, WA, USA) and IBM's Statistical Package for Social Sciences (SPSS, Ver. 28.0). Performance metrics of the CNN include sensitivity, specificity, positive and negative
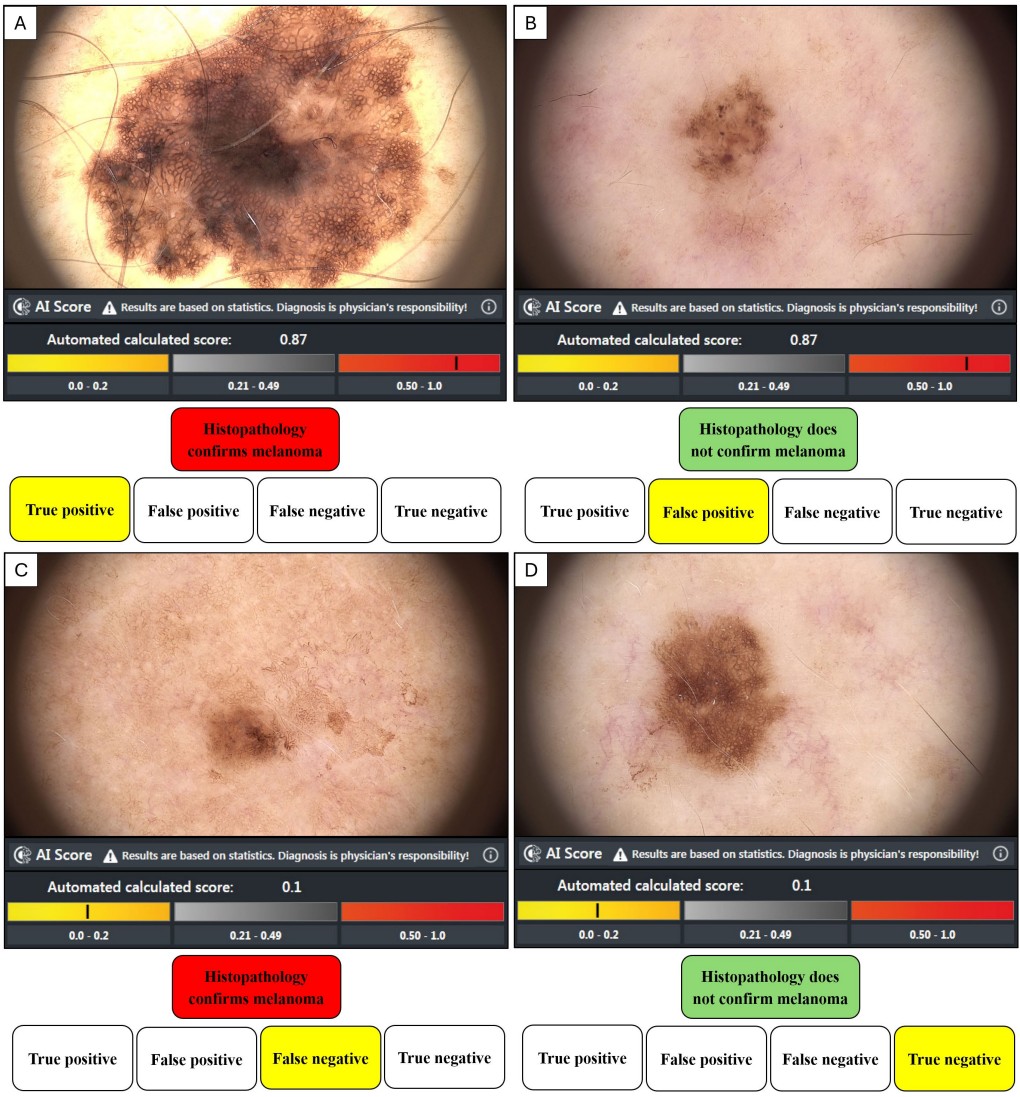

**Figure 3 The dichotomous relationship between histological analysis and AI risk score.** (A) True positive when histopathology confirms a melanoma diagnosis and AI score highlights suspicion. (B) False positive when histopathology does not confirm a melanoma diagnosis, but the AI score suggests the lesion is suspicious. (C) False negative when histopathology confirms a melanoma diagnosis, but the AI score suggests the lesion is unsuspicious. (D) True negative when histopathology does not confirm a melanoma diagnosis and the AI score suggests the lesion was unsuspicious.

predictive value, and diagnostic odd ratio. The area under the curve (AUC) of the receiver operating characteristic curves (ROC) was used to assess the accuracy of the CNN for the classification of images for each clinic. Chi-squared tests and independent sample $t$-tests were used to compare the two clinics. Alpha was predetermined to be significant between the two clinics at $p < 0.05$.

**Table 2** Patient demographics, including number of participants (n) and corresponding percentage (%).

| Characteristics | Overall, *n* (%) | Gold coast, *n* (%) | Townsville, *n* (%) | *p*-value |
|---|---|---|---|---|
| Participants, *n* (%) | 325 (100) | 180 (55.4) | 145 (44.6) | <0.001 |
| Age, years [±SD] | 54.8 [15.7] | 53.4 [14.4] | 56.5 [17.1] | 0.087 |
| Gender, *n* (%) | | | | 0.859 |
|    Male | 191 (58.8) | 105 (58.3) | 86 (59.3) | |
|    Female | 134 (41.2) | 75 (41.7) | 59 (40.7) | |
| Positive Family History of MM, *n* (%) | 70 (21.5) | 58 (32.2) | 12 (8.3) | <0.001 |
| Previous History of MM, *n* (%) | 76 (23.4) | 35 (19.4) | 41 (28.3) | 0.062 |
| Queried lesions, *n* (%) | 373 (100) | 214 (57.4) | 159 (42.6) | <0.001 |
| Queried lesions grouped by Gender, *n* (%) | | | | 0.887 |
|    Male | 226 (60.6) | 129 (60.3) | 97 (61.0) | |
|    Female | 147 (39.4) | 85 (39.7) | 62 (39.0) | |
| Melanoma Total, *n* (%) | 114 (100) | 78 (68.4) | 36 (31.6) | 0.592 |
|    *In situ* | 85 (74.6) | 57 (67.1) | 28 (32.9) | |
|    Invasive | 29 (25.4) | 21 (72.4) | 8 (27.6) | |

# RESULTS

## Study participants

There was a total of 325 participants (180 from Gold Coast and 145 from Townsville) who volunteered to participate in this study. The average age was 54.8 years (53.4 Gold Coast, 56.5 Townsville, $p = 0.87$) and most of the participants were male (60.5% male, 39.5% female). The majority of patients had a Fitzpatrick skin type of type I–III (inclusive), with one patient in Townsville having a Fitzpatrick skin type of IV. For Gold Coast participants, approximately one-third (32.2%) of participants had a previous family history of MM, and 22.9% of participants had a previous MM diagnosis. The participants in Townsville had a lower percentage regarding positive family history of MM (8.3%, $p < 0.001$) and a larger percentage of patients with a prior history of MM (28.3%, $p = .062$). In total, there were 373 queried lesions identified during the screening in this study ($n = 214$ Gold Coast, $n = 159$ Townsville, $p < 0.001$). Of these queried lesions, histopathology confirmed 114 were MM ($n = 79$ Gold Coast, $n = 35$ Townsville), with 85 *in situ* ($n = 58$ Gold Coast, $n = 27$ Townsville) and 29 invasive ($n = 21$ Gold Coast, $n = 8$ Townsville) (Table 2).

## Performance metrics of CNN

Regarding combined results from both clinics, the CNN performance metrics on lesions investigated by histopathology, sensitivity was calculated to be 63.2% (61.5% Gold Coast, 66.7% Townsville), specificity was 53.7% (55.1% Gold Coast, 52.0% Townsville), positive predictive value was 37.8% (44.0% Gold Coast, 28.9% Townsville) and negative predictive value was 76.8% (71.4% Gold Coast, 84.2% Townsville). Both the positive and negative likelihood ratios (LR+ and LR-) and diagnostic ratios (DOR) did not differ between either of the two clinics and were 1.4 (LR+), 0.7 (LR-) and 2.0 (DOR), respectively. The overall accuracy was 56.6% (57.5% Gold Coast, 56.1% Townsville) (Table 3).

**Table 3 Performance metrics of CNN between the clinics.**

| Clinic | Sen/TPR (%) | Spe/TNR (%) | Acc (%) | Prev (%) | PPV (%) | NPV (%) | FPR (%) | FNR (%) | LR+ | LR- | DOR |
|---|---|---|---|---|---|---|---|---|---|---|---|
| Gold Coast (CNN) | 61.5 | 55.1 | 57.5 | 50.9 | 44.0 | 71.4 | 44.9 | 38.5 | 1.4 | 0.7 | 2.0 |
| Towns-ville (CNN) | 66.7 | 52.0 | 56.1 | 55.3 | 28.9 | 84.2 | 48.4 | 34.3 | 1.4 | 0.7 | 2.0 |
| Total Clinics (CNN) | 63.2 | 53.7 | 56.6 | 51.5 | 37.8 | 76.8 | 46.3 | 36.8 | 1.4 | 0.7 | 2.0 |

Notes.

Where:

Sen, Sensitivity; Spe, Specificity; Acc, Accuracy; Prev, Prevalence; TPR, True positive rate; TNR, True negative rate; PPV, Positive predictive value; NPV, negative predictive value; FPR, False positive rate; FNR, False negative rate; LR+, positive likelihood ratio; LR-, negative likelihood ratio; DOR, Diagnostic odds ratio; CNN, Convolutional neural network.

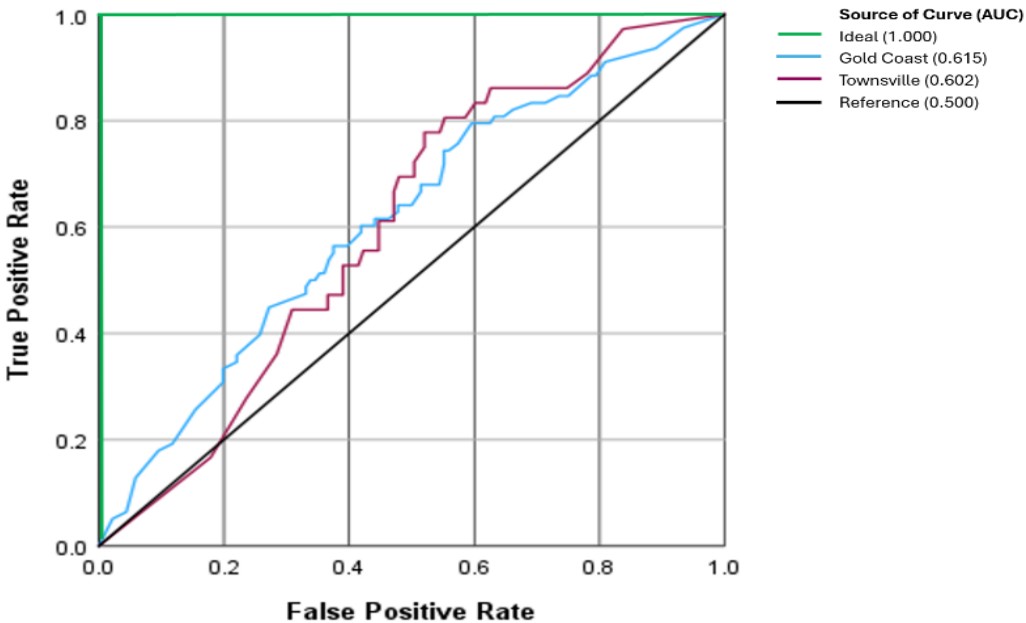

**Figure 4 Receiver operator characteristic curve between clinics.** The ideal AUC is the theoretical perfect result with 100% accuracy (AUC = 1.0). The Reference AUC represents the ratio of results happening by chance (AUC = 0.5).

The performance as observed on the receiver operating characteristics curve (ROC), the area under the curve (AUC), a measure of accuracy, was 0.615 for Gold Coast and 0.602 for Townsville ($p = 0.834$), respectively (Fig. 4).

Combining the histopathology from Gold Coast and Townsville yielded 85 *in situ* and 29 invasive MM. There were 72 MM that the CNN highlighted as suspicious (*in situ* 60%, invasive 72.4%, $p = 0.231$) and 42 lesions with a score that was not suspicious of malignancy (Table 4).

## Comparison of CNN performance between clinics

When comparing the performance of the CNN between clinics for MM, based upon the manufacturer's categorisation, there was no significant difference for either melanoma ($p = 0.598$) or non-MM ($p = 0.616$). The percentage of TP (histopathology-confirmed

**Table 4  Comparison of CNN diagnostic separated as *in situ* and invasive melanoma.**

| Combined clinics | *In-situ* n (%) | Invasive n (%) | Total n (%) |
|---|---|---|---|
| 0.50 or greater | 51 (60.0) | 21 (72.4) | 72 (63.2) |
| 0.49 or lower | 34 (40.0) | 8 (27.6) | 42 (36.8) |
| Total | 85 (74.6) | 29 (25.4) | 114 (100) |

**Table 5  Performance of CNN compared to histopathology analysis.**

| Clinic | Characteristics of CNN | Melanoma n (%) | Not Melanoma n (%) | Total n (%) |
|---|---|---|---|---|
| | 0.50 or greater | 48 (61.5) | 61 (44.9) | 109 (50.9) |
| Gold Coast | 0.49 or lower | 30 (38.5) | 75 (55.1) | 105 (49.1) |
| | Total | 78 (100) | 136 (100) | 214 (100) |
| | 0.50 or greater | 24 (66.7) | 59 (48.0) | 83 (52.2) |
| Townsville | 0.49 or lower | 12 (33.3) | 64 (52.0) | 76 (47.8) |
| | Total | 36 (100) | 123 (100) | 159 (100) |
| | 0.50 or greater | 72 (63.2) | 120 (46.3) | 192 (51.5) |
| Combined | 0.49 or lower | 42 (36.8) | 139 (53.7) | 181 (48.5) |
| | Total | 114 (100) | 259 (100) | 373 (100) |

MM) was similar between both clinics (61.5% Gold Coast, 66.7% Townsville). Similarly, the percentage of TN was close between clinics (non-MM as confirmed by histopathology) (49.1% Gold Coast, 47.8% Townsville). With regards to the FN percentage (CNN classified negative; however, histopathology confirmed MM), it was also similar between both clinics (38.5% Gold Coast, 33.3% Townsville). Additionally, the percentage of FP (CNN classified positive; however, histopathology was not confirmed as a MM) was also similar between both clinics (50.9% Gold Coast, 52.2% Townsville) (Table 5).

## DISCUSSION

Skin cancer identification is the perfect opportunity to test AI in a medical setting as it is simple to use, quick and its non-invasive nature has the potential to limit unnecessary biopsies, reducing morbidity for patients. Performance of AI for MM detection has been favourable (*Miller et al., 2024*). However, many publications are limited to image datasets that do not necessarily reflect accurate performance within a medical setting. It was the aim of this study to report on the performance of a market-approved CNN to differentiate lesions suspicious of MM from benign lesions, and to compare this performance across two separate skin cancer-focused clinics within Australia.

Previously, our group had investigated incorporating AI in screening for MM in a primary care setting, with a reported accuracy of 54.2% (*Miller et al., 2023*). Limitations included investigating a relatively small sample size ($n = 48$) of suspicious MM lesions and potential selection bias as a single clinician was identifying lesions for investigation. The present study included four clinicians who operated independently and a larger sample size ($n = 373$), which resulted in a similar combined accuracy of 56.6%.

We identified that the sensitivity of the CNN technology utilised in this study at both clinical data collection sites (clinics combined, 63.2%) had improved slightly from our previously reported findings (53.3%) (*Miller et al., 2023*). However, when we compared the specificity to our previous study, we found the specificity remained comparable (53.9% current study, 54.4% previous findings). In an additional study reporting clinical findings within Australia, *Menzies et al. (2023)* reported two CNN-based technologies with modest sensitivity (16.4%, 50.9%) values in contrast to higher specificity (94.0%, 98.3%) values. Accuracy for both the CNNs investigated in the study *Menzies et al. (2023)* was calculated as 72.1% and 80.2%, respectively, as reported in our recent systematic review, which compared market-approved CNN technology in the classification of MM (*Miller et al., 2024*).

The performance metrics reported by studies located in Australia (*Miller et al., 2023*; *Menzies et al., 2023*) are lower than those of studies conducted in American (*MacLellan et al., 2021*) and European-based locations (*Winkler et al., 2023*; *Fink et al., 2020*; *Winkler et al., 2019*; *Winkler et al., 2021b*; *Winkler et al., 2022*) using the same commercial device utilised in this study. For example, sensitivity for MM in Australia (including this current study) ranged from 53.3–63.2% compared with 81.6–97.1% in the American/European studies. Specificity varied greatly between continents, with Australian studies reporting 53.9–54.4% compared to 78.8–88.9% in the American and European studies. Accuracy was also reported to be lower, ranging from 54.4–56.7% in Australia and 81.3–87.7% in America/Europe, respectively. Varying participant numbers between the studies make direct comparisons problematic. However, the variation in performance metrics of market-approved CNN clearly highlights a notable difference between Australia and studies published in other locations globally.

We found no statistically significant difference in CNN accuracy between *in situ* and invasive MM in the participants of this study. This is a significant finding as invasive MM often demonstrates more definitive dermatoscopic features that we expect the CNN would classify as a suspicious lesion. Though the difference in sensitivity of the CNN was not statistically significant, there may be clinical relevance, as there was a 12.4% improvement in sensitivity between *in situ* and invasive MM. Two studies have investigated CNN discrimination between invasive and *in situ* MM, with both studies reporting greater performance for lesions where the Breslow thickness was >1.0 mm (*Gillstedt et al., 2022*; *Polesie et al., 2021*). Both studies reported lower CNN performance when the depth of MM invasion was <1.0 mm. We must acknowledge the relatively small sample size of invasive MM of our study ($n = 29$) and all of these malignancies diagnosed did not exceed 1.0 mm. We hypothesize that the investigated CNN is likely to perform better when the Breslow thickness exceeds 1.0 mm in depth. However, a higher sample size would be necessary to test such a hypothesis in our population.

In dichotomous analysis (*i.e.*, only two possible values), such as in our study, positive predictive value (PPV) and negative predictive value (NPV) indicate how well the classification system performed. The values from our combined clinics suggest poorer performance for PPV and greater performance for NPV. Put simply, CNN is less likely to correctly predict MM when lesions are suggested as suspicious and more likely to regard

non-suspect lesions as unsuspicious. However, PPV and NPV may be influenced if the population numbers are not even, as evident in our investigated population group (uneven MM numbers compared to non-MM). A greater prevalence of MM will positively influence PPV and negatively influence NPV. This is observed in our dataset with a greater prevalence of MM diagnosed in Gold Coast compared to Townsville.

Likelihood ratios (LR+ and LR-) and diagnostic odds ratios (DOR) consider uneven prevalence and may be more appropriate outcome measures. Clinical relevance of likelihood ratios has been suggested to be >10 (LR+) and <0.1 (LR-) (*Caraguel & Colling, 2021*; *Hayden & Brown, 1999*). For reference, a DOR of "1" suggests no change or any change happening by chance. The performance of the CNN on our study population was 1.4 (LR+) and 0.7 (LR-), respectively. Given these DORs, there was no discernible difference between the two clinics, which suggests selection bias was not dependent upon location or clinic.

The findings of our present study and previous studies by *Miller et al. (2023)* and *Menzies et al. (2023)* provide evidence to suggest that a master image repository upon which the CNN is trained may not be appropriate for all skin types. This echoes previous comments suggesting that the development of separate datasets for differing populations may be more beneficial than one diverse algorithm (*Adamson & Smith, 2018*). Due to the proprietary algorithm of the CNN utilised in this study, we are unsure what degree skin type, if at all, would have on the performance metrics and, subsequently, the predictive score of the market-approved algorithm. Currently, there remains a lack of transparency and underrepresentation of images pertaining to Fitzpatrick skin types IV–VI in publicly available datasets associated with pigmented skin lesions (*Liu et al., 2023*).

In this study, the reported AUC of 0.602–0.615 is indicative of average diagnostic ability (*Nahm, 2022*), meaning the CNN performed slightly better than chance in distinguishing between MM and non-MM lesions. For reference, an AUC of 0.5 indicates no discrimination, whereas an AUC of 1.0 has perfect discrimination (*Nahm, 2022*). The impact of an "average" AUC is a high rate of false negatives and false positives, or malignant lesions missed and benign lesions unnecessarily excised. The anticipated outcomes are increased morbidity, unnecessary anxiety, additional testing, delayed diagnosis, and missed treatments. Therefore, enhancing the CNN's performance is critical for effective skin cancer management.

The strength of this study includes the use of images of multiple real-world clinical practices. Though there may be an argument to suggest that a sample size of 373 queried lesions remains modest, the rate of data accumulation in general practice is gradual, reflecting the scenario (or chance) that a patient has an MM on their skin. The incorporation of multiple clinicians identifying lesions reduces the impact of selection bias for this study. Additionally, all MM were confirmed *via* histopathology, recognised as the gold standard for MM determination (*Melanoma Research Alliance, 2024*).

There are limitations to this study. At the time of the inception of this study, our group had started with one clinician on the Gold Coast, Queensland. Several months later, the study was expanded to include several other clinicians in Townsville, Queensland. This, in part, explains the uneven data collection between the two clinics. This difference in lesion numbers may, in part, explain the difference in sensitivity we have reported between the

two clinics of this study. Though the performance of the metrics reported in this study is less than favourable, this does not negate the valuable service a bedside CNN can provide a clinician for the assessment and management of melanocytic lesions of concern. At the current time, our group does not have access to a comparison technology to assess how it may perform. As such, the market-approved CNN's performance was limited to one commercially available CNN. Finally, the limitation of the HD camera scope meant larger lesions beyond 40 mm were excluded.

We acknowledge the possibility that individual patients might have contributed multiple lesions to this study. However, each lesion was analysed independently as per current diagnostic accuracy study conventions. Future research could explore the impact of potential intra-patient correlation through advanced statistical methods such as mixed-effect modelling.

The algorithm tested in this study is a commercially available product, and the company retains a proprietary license over this property. As a result, there is limited information publicly available pertaining to both the architecture of the CNN used as well as the image dataset employed to train it on. The prototype for the algorithm is based on Google's CNN architecture, Inception v4 (*Szegedy et al., 2016*), first described in 'Man against machine' (*Haenssle et al., 2018*) and then further by 'Man against machine reloaded' (*Haenssle et al., 2020*) as a supplementary file. The dataset used to train the CNN has previously been reported as >150,000 labelled dermatoscopic images (*Winkler et al., 2021a*). Despite the proprietary nature of the algorithm, the Inception v4 architecture is documented and validated in medical imaging literature, providing general confidence in our approach (*Emara et al., 2019*). However, further transparency from manufacturers would significantly enhance interpretability and reduce potential biases in clinical implementation.

## CONCLUSION

In conclusion, we advise that diagnostic accuracy for MM remains an issue for market-approved CNN when performing in primary care settings in Australia. Further review of the image sets the CNN is trained on may be necessary, as MM features and stage of diagnosis may differ between countries.

## ACKNOWLEDGEMENTS

We would also like to extend our sincere thanks to Professor Pat O'Shea, friend and mentor, for instilling a passion for research; you are sincerely missed but not forgotten.

### Funding

This study was funded by Johnson and Johnson for the dermatoscope that included AI. An Australian Government research training scholarship provides financial support to

lead author Ian Miller for his PhD studies. The funders had no role in study design, data collection and analysis, decision to publish, or preparation of the manuscript.

### Grant Disclosures
The following grant information was disclosed by the authors:
Johnson and Johnson.
Australian Government.

### Competing Interests
Mike Climstein is a Section Editor for PeerJ (Sports Medicine and Rehabilitation). Michael Stapelberg is employed by Skin Clinic Robina. Jeremy Hudson, Paul Coxon, and Nathaniel Milani are employed by North Queensland Skin Cancer Centre. Joe Walsh is employed by Sport Science Institute and AI Consulting Group.

### Author Contributions
- Ian J. Miller conceived and designed the experiments, performed the experiments, analyzed the data, prepared figures and/or tables, authored or reviewed drafts of the article, and approved the final draft.
- Michael Stapelberg conceived and designed the experiments, performed the experiments, analyzed the data, authored or reviewed drafts of the article, and approved the final draft.
- Jeremy Hudson conceived and designed the experiments, performed the experiments, analyzed the data, authored or reviewed drafts of the article, and approved the final draft.
- Paul Coxon performed the experiments, analyzed the data, authored or reviewed drafts of the article, and approved the final draft.
- Nathaniel Milani performed the experiments, analyzed the data, authored or reviewed drafts of the article, and approved the final draft.
- Nedeljka Rosic conceived and designed the experiments, analyzed the data, prepared figures and/or tables, authored or reviewed drafts of the article, and approved the final draft.
- James Furness analyzed the data, authored or reviewed drafts of the article, and approved the final draft.
- Joe Walsh analyzed the data, authored or reviewed drafts of the article, and approved the final draft.
- Mike Climstein conceived and designed the experiments, analyzed the data, prepared figures and/or tables, authored or reviewed drafts of the article, and approved the final draft.

### Human Ethics
The following information was supplied relating to ethical approvals (*i.e.*, approving body and any reference numbers):
This study was granted approval by the Southern Cross University's Human Research Ethics Committee (11th May 2020/47).

## Data Availability

Raw data is available in the Supplemental Files.

## Supplemental Information

Supplemental information for this article can be found online at http://dx.doi.org/10.7717/peerj.19876#supplemental-information.

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
