# Peer review of "Market-approved convolutional neural network tasked with classifying skin lesions under suspicion of melanoma: performance across primary care clinics within Australia"

_PeerJ, doi:10.7717/peerj.19876_

## Round 0.1 · original submission · Major Revisions

While your study addresses an important area of research, the reviewers and I have identified a few areas that require major revisions. The following points must be addressed thoroughly in your revised manuscript:

Transparency of the CNN algorithm:

The manuscript relies on a "market-approved" convolutional neural network (Moleanalyzer-Pro) for the classification of skin lesions. However, you provide insufficient detail regarding the algorithm's architecture, training data, and preprocessing steps. This lack of transparency is a major concern. Why was it selected? Are there other commercial apps?
We require a more detailed discussion of the limitations imposed by the proprietary nature of the algorithm. You must explicitly address how this lack of information hinders the assessment of potential biases and the interpretation of your results, particularly concerning the impact of skin type variations.


Comparative Analysis and Geographic Heterogeneity:
While the discussion acknowledges the variability of results across different geographic locations, it fails to adequately explore the underlying factors contributing to these discrepancies. A significantly expanded discussion is required, delving into potential variables such as variations in patient populations, imaging techniques, and the specific configurations of the convolutional neural network (CNN) algorithms employed. This analysis should attempt to identify and discuss potential explanations, including differences in melanoma prevalence and characteristics, variations in CNN training datasets, potential differences in image acquisition and preprocessing techniques, and disparities in the Fitzpatrick skin types represented in the datasets.

These revisions are essential to ensuring the rigor and clarity of your manuscript. We believe that addressing these concerns will significantly strengthen your study and enhance its contribution to the field.

Please submit a revised manuscript that addresses these points comprehensively, along with a detailed point-by-point response to this letter and the reviewers' comments.

Reviewer 1 ·

Basic reporting

Thank you for this interesting paper comparing a market-approved CNN on a selected Australian primary care population for the discrimination of malignant melanoma.

Overall the paper is well written with sound background, referencing, structured methodologies and results.

I have some points that would need to be addressed to make the paper even clearer for the intended audience/readers.

Experimental design

1. The prevalence of MM in the selected study population (surfers) is higher than the average population prevalence and should be clarified even more as it affects several primary outcomes of the study

2. The ethics statement and the informed consent form do not contain sufficient information to be properly reviewed. Please add the Patient Information Sheet that details the written (or oral) information given to the presumptive participants.

3. Table 3 is missing a column, if not please provide some more clarification to the first column.

4. An a priori statistical calculation (primary hypothesis and power calculation are missing). Can you clarify the convenience sampling method from 2021-2024 and why inclusion was concluded at 325 participants (372 lesions)?

5. What were the experience levels (clinical/dermatological/dermatoscopical etc.) of the four recruiting doctors/GPs? Please provide a small demographic table in an appendix section, or add a paragraph about it in methods.

Validity of the findings

1. The article body is missing a COI statement and a Financial statement (external funding from Johnson&Johnson).

2. The Sens and Spec (and other performance metrics) from the participating doctors could be added as discussion material in the article to put things in a wider context. Was the CNN device intended to be autonomous or augment the GPs performance?

3. Provide some reasoning about selection of lesions and the lesions/participants excluded from the study as "not suspected to be a melanoma" - potential false negatives. All 372 lesions were "labelled" as potential melanomas from the GPs.

4. Comment on the resulting PPV/NPV between the two sites in relation to the prevalence of MM and the selection criteria/GPs experience.

Reviewer 2 ·

Basic reporting

Here some recommendations for the authors regarding basic reporting:

1. I think it would be appropriate to include the secondary aim of comparing the performance of the CNN algorithm/software between the centers in the abstract.
2. Please reframe the aims on line 32-36 in a more organized way. [e.g. we aimed 1) to investigate .... and 2) to compare .....]
3. For the methods, I think it might be appropriate to mention that this is a "diagnostic accuracy study"
4. I think the authors can reframe what was mentioned in lines 95-86 as "Average age was *** and most of participants were male (percent, n)."
5. Despite the low skin of color representation was mentioned, I recommend the authors to elaborate more on how this might affect the behaviour of the system and what are the implications of this low representation in the study
6. I recommend the authors to change the ordering of the results. I would start first talking about the performance of the CNN in the two clinics (overall performance), and then, in a section after that, showing the results stratified by clinic.
7. In line 141-143, please replace "accuracy" by "performance"
8. Please include the appropriate citations in lines 170-174 regarding the studies from Europe and US

Experimental design

Please see some questions/recommendations to the authors regarding the methods and experimental design:

1, Was a power analysis performed as part of the study? If that is the case, please include a summary in the methods section.
2. Was histopathology evaluated by one provider? Please specify if this provider is a pathologist or a dermatopathologist.
3. I recommend to the authors to include additional information on the software/device/CNN algorithm used. Maybe this can be part of a supplementary material so readers can have better insights on the properties of the system.
4. I recommend that the information included in the limitations about the size of the lesions also be listed as an exclusion criteria in the methods.
5. I think it would be appropriate to elaborate more on how was the data capturing process. Was any quality assessment process done for the images included in the study and assessed by the CNN?

Validity of the findings

Please see in the following paragraphs, some recommendations and questions to the authors regarding the validity of the findings and results:

1. It is not 100% clear if all participants participated with just one lesion and one image or if one participant contributed with multiple lesions/images. This is relevant as the use of non-independent data would dictate if there is potential existence of additional relationships/correlation between data observations that would necessitate consideration of specific statistical tools for correlated data/repeated measurements.
2. Despite that the authors mentioned that there was not a significant difference between the performance of the system in the two clinics, some metrics showed a difference of at least 5% like the case of the percentage of true positives. Some readers might find this difference as clinically relevant, so I think it would be appropriate to elaborate more on this in the discussion.
3. Again, similar to the previous comment, the authors mentioned that there was not a statistically significance difference in detection of in-situ and invasive MM, but the difference of proportions was of around 12% (without considering a confidence interval), which might be clinically relevant and might suggest a potential issue with the sample size (power) as the statistical test showed a high p value. I recommend the authors to focus more on the clinical relevance of this finding, rather than solely on the statistical significance, and to include along this analysis that there is considerable uncertainty surounding this result as per the limited sample size of the study.
4. I recommend the authors to discuss in the limitations section the implications of performing this study on lesions with high pretest probability (e.g., pigmented lesions already assessed by clinicians as potentially MM) and how that affects the interpretation of the results. Additionally, I wonder if in the studies evaluating this same system in US or Europe the inclusion/exclusion criteria, as well as, the participant screening/recruitment process was similar.

Additional comments

The manuscript is well written and easy to navigate. All my suggestions were included in the previous sections.

Overall, I would like to congratulate the authors for conducting this research, as this data contribute to the understanding of the performance of AI systems in real clinical scenarios. Despite the limitations of the study, and the sample size, I consider there are great and valuable contributions from this research. However, from my point of view, revisions and modifications are necessary before publication.

---

## Round 0.2 · Minor Revisions

Thank you for your revised submission and for your openness and readiness to address the reviewers’ comments. We appreciate the effort you have made in improving the manuscript. The revised version is much improved. However, a few minor revisions are still required before the manuscript can be accepted for publication.

One of the reviewers has provided additional comments that we believe will further strengthen the paper: Furthermore, below are some points that need to be addressed:

In the updated version, you mention that the physician may, upon request, reveal the CNN results. It would be helpful to clarify how many physicians actually did so in practice—specifically, in cases where they judged the lesion to be suspicious and recommended a biopsy. This detail would enhance the understanding of how the AI tool influenced clinical decision-making.

The discussion section could be more concise and focused. Rather than reiterating each result, we encourage you to concentrate more on interpreting your findings. In particular, consider discussing them in the broader context of using AI to support diagnostic accuracy, especially in the detection of malignant melanoma (MM).

Additionally, the beginning of the discussion could be strengthened by including a brief introduction or summary of the study's aims. This would provide a smoother transition into the discussion of your previous findings and how this study builds upon them.

Reviewer 1 ·

Basic reporting

Thank you for a clear and well-motivated revision of your manuscript. The authors' responses to the reviewers' and editor's comments have, in my opinion, been fully met. I recommend publication of the revised version of the manuscript.

Experimental design

no comment

Validity of the findings

no comment

Reviewer 2 ·

Basic reporting

I recommend reviewing the following details in the manuscript:

1. Please consider modifying the phrases in lines 199-201 and 207-208 to improve clarity. Here my recommendations for your reference:

"Though the (difference in) sensitivity of (the) CNN was not statistically significant, there may still be some clinical relevance as there was a 12.4% improvement in sensitivity between in situ and invasive MM. Two studies have investigated CNN discrimination between invasive and in situ
MM, with both studies reporting greater performance for lesions where the Breslow thickness was >1.0mm (20, 21). Both studies reported lower CNN performance when the depth of MM invasion was <1.0mm. We must acknowledge the relatively small sample size of invasive MM of our study (n=29) and all of these malignancies diagnosed did not eclipse 1.0mm. We hypothesize that the investigated CNN is likely to perform better when the Breslow thickness exceeds 1.0mm in depth. However, (a higher sample size would be necessary to test such hypothesis in our population.)"

2. Please check the phrases on lines 228-230. It seems that there is a missing word:

"Due to the proprietary algorithm of the CNN utilised in this study, we are unsure to what degree skin type, if at all, would have on the performance metrics and, subsequently, the predictive score of the market-approved algorithm."

3. Please check figure 3 caption. Specifically, please correct the definition of "False positive" in part (B):

"Figure 3
The dichotomous relationship between histological analysis and AI risk score.
'(A) True positive when histopathology confirrmed melanoma and AI score highlighted suspicion. (B) False positive when histopathology confirmed melanoma but AI score suggested lesion is unsuspicious. (C) False negative when histopathology confirmed melanoma but AI score suggested lesion is unsuspicious. (D) True negative when histopathology did not confirm melanoma and the AI score suggested the lesion was
unsuspicious.'"

4. Please check the formating of table 4. The"combined clinics" row is empty as the total or combined values are in the "total" row.

Experimental design

1. In the "Response to reviewers" it was stated that no lesions were excluded/omited based on size. However, in lines 256-257 of the manuscript it is specified that larger lesions beyond 40mm were excluded as per the limitations of the scanner. Please clarify.

Validity of the findings

1. The "Response to reviewers" state that t-tests were used for the comparison of diagnostic performance. However, technically speaking, values such as sensitivity, specificity, PPV, NPV, are proportions, for which other set of statistical tests are more appropriate (e.g., test for two proportions, chi-square, etc: https://www.ncss.com/wp-content/themes/ncss/pdf/Procedures/PASS/Tests_for_Two_Independent_Sensitivities.pdf). Could you please specify if the appropriate set of tests were used to compare those metrics? Also, please confirm that the appropriate statistical methods were used to compare AUC-ROCs.

2. Please clarify the meaning of "Queried lesions by gender" reported in table 2.

Additional comments

The authors did a fantastic job solving most of the comments from the previous review, providing solid evidence of the validity of their research. However, a couple set of details need to be clarified/addressed/corrected before final publication.

---

## Round 0.3 · Minor Revisions

Thank you for submitting your revised manuscript and for addressing the reviewers' comments promptly. We appreciate your efforts in improving the manuscript.

The reviewers have re-evaluated your submission, and their comments are appended below. We ask that you carefully consider and address each point raised.

In addition to the reviewer feedback, I have one further comment regarding the methodology. It is still not entirely clear how physicians could have checked the algorithm's response in your proposed workflow. Thank you for the initial clarification but to ensure complete understanding of the methodology, could you please elaborate on the exact workflow regarding the CNN results? We're particularly interested in the precise timing of when clinicians had the option to view these results relative to their independent biopsy decisions, and if any data was collected on the frequency of such access, particularly for lesions deemed suspicious and recommended for biopsy.

Reviewer 2 ·

Basic reporting

Please check (4) additional comments.

Experimental design

Please check (4) additional comments.

Validity of the findings

Please check (4) additional comments.

Additional comments

I would like to thank you the authors for their treamendous work answering to all reviewers comments.
Before publication, I just want to confirm the following two details with the authors, as were not completely clear for me in the "authors response letter":
1. The authors clarified that they used the appropriate statistical methods to compared proportions (performance metrics). However, no clarification on if whether or not the appropriate statistical methods were used to compare AUC-ROCs were used. Please clarify and correct as appropriate.
2. Please clarify the meaning of the label "Queried lesions grouped by Gender" and how this differ from the field on "Gender" in table 2.

---

## Round 0.4 · accepted · Accept

Thank you for your revised submission. I am pleased to inform you that your manuscript has been accepted for publication. We appreciate your patience and the thoughtful revisions that have strengthened the work.